# Feature Interaction Interpretability: A Case for Explaining Ad-Recommendation Systems via Neural Interaction Detection

**Michael Tsang**[1], **Dehua Cheng**[2], **Hanpeng Liu**[1], **Xue Feng**[2], **Eric Zhou**[2], and **Yan Liu**[1]

[1]Department of Computer Science, University of Southern California
{tsangm,hanpengl,yanliu.cs}@usc.edu
[2]Facebook AI
{dehuacheng,xfeng,hanningz}@fb.com

## ABSTRACT

Recommendation is a prevalent application of machine learning that affects many users; therefore, it is important for recommender models to be accurate and interpretable. In this work, we propose a method to both interpret and augment the predictions of black-box recommender systems. In particular, we propose to interpret feature interactions from a source recommender model and explicitly encode these interactions in a target recommender model, where both source and target models are black-boxes. By not assuming the structure of the recommender system, our approach can be used in general settings. In our experiments, we focus on a prominent use of machine learning recommendation: ad-click prediction. We found that our interaction interpretations are both informative and predictive, e.g., significantly outperforming existing recommender models. What's more, the same approach to interpret interactions can provide new insights into domains even beyond recommendation, such as text and image classification.

## 1 INTRODUCTION

Despite their impact on users, state-of-the-art recommender systems are becoming increasingly inscrutable. For example, the models that predict if a user will click on an online advertisement are often based on function approximators that contain complex components in order to achieve optimal recommendation accuracy. The complex components come in the form of modules for better learning relationships among features, such as interactions between user and ad features (Cheng et al., 2016; Guo et al., 2017; Wang et al., 2017; Lian et al., 2018; Song et al., 2018). Although efforts have been made to understand the feature relationships, there is still no method that can interpret the feature interactions learned by a generic recommender system, nor is there a strong commercial incentive to do so.

In this work, we identify and leverage feature interactions that represent how a recommender system generally behaves. We propose a novel approach, Global Interaction Detection and Encoding for Recommendation (GLIDER), which detects feature interactions that span globally across multiple data-instances from a source recommender model, then explicitly encodes the interactions in a target recommender model, both of which can be black-boxes. GLIDER achieves this by first utilizing our ongoing work on Neural Interaction Detection (NID) (Tsang et al., 2017) with a data-instance perturbation method called LIME (Ribeiro et al., 2016) over a batch of data samples. GLIDER then explicitly encodes the collected global interactions into a target model via sparse feature crossing.

In our experiments on ad-click recommendation, we found that the interpretations generated by GLIDER are illuminating, and the detected global interactions can significantly improve the target model's prediction performance. Because our interaction interpretation method is very general, we also show that the interpretations are informative in other domains: text, image, graph, and dna modeling.

---

Code is available at: https://github.com/mtsang/interaction_interpretability

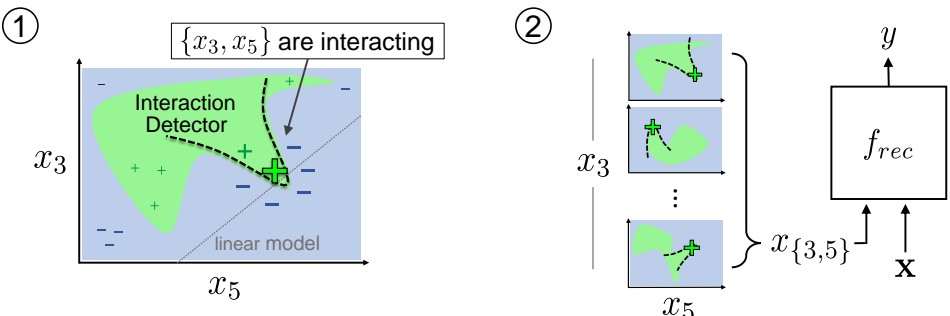

Figure 1: A simplified overview of GLIDER. ① GLIDER utilizes Neural Interaction Detection and LIME together to interpret feature interactions learned by a source black-box model at a data instance, denoted by the large green plus sign. ② GLIDER identifies interactions that consistently appear over multiple data samples, then explicitly encodes these interactions in a target black-box recommender model $f_{rec}$.

Our contributions are as follows:

1. We propose feature interaction interpretations of general prediction models via interaction detection.

2. Based on this approach, we propose GLIDER to detect and explicitly encode global feature interactions in black-box recommender systems. This process is a form of automatic feature engineering.

3. Through experiments, we demonstrate the overall interpretability of detected feature interactions on a variety of domains and show that the interactions can be leveraged to improve recommendation accuracy.

## 2 NOTATIONS AND BACKGROUND

**Notations**: Vectors are represented by boldface lowercase letters, such as $\mathbf{x}$ or $\mathbf{z}$. The $i$-th entry of a vector $\mathbf{x}$ is denoted by $x_i$. For a set $\mathcal{S}$, its cardinality is denoted by $|\mathcal{S}|$.

Let $d$ be the number of features in a dataset. An *interaction*, $\mathcal{I}$, is a subset of feature indices: $\mathcal{I} \subseteq \{1, 2, \ldots, d\}$, where $|\mathcal{I}|$ is always $\geq 2$. A higher-order interaction always has $|\mathcal{I}| \geq 3$. For a vector $\mathbf{x} \in \mathbb{R}^d$, let $\mathbf{x}_{\mathcal{I}} \in \mathbb{R}^{|\mathcal{I}|}$ be restricted to the dimensions of $\mathbf{x}$ specified by $\mathcal{I}$.

Let a black-box model be $f(\cdot) : \mathbb{R}^p \to \mathbb{R}$. A black-box recommender model uses tabular feature types, as discussed later in this section. In classification tasks, we assume $f$ is a class logit. $p$ and $d$ may be different depending on feature transformations.

**Feature Interactions**: By definition, a model $f$ learns a statistical (non-additive) feature interaction $\mathcal{I}$ if and only if $f$ cannot be decomposed into a sum of $|\mathcal{I}|$ arbitrary subfunctions $f_i$, each excluding a corresponding interaction variable (Friedman et al., 2008; Sorokina et al., 2008; Tsang et al., 2017), i.e., $f(\mathbf{x}) \neq \sum_{i \in \mathcal{I}} f_i(\mathbf{x}_{\{1,2,\ldots,d\}\setminus i})$.

For example, a multiplication between two features, $x_1$ and $x_2$, is a feature interaction because it cannot be represented as an addition of univariate functions, i.e., $x_1 x_2 \neq f_1(x_2) + f_2(x_1)$.

**Recommendation Systems**: A recommender system, $f_{rec}(\cdot)$, is a model of two feature types: dense numerical features and sparse categorical features. Since the one-hot encoding of categorical feature $x_c$ can be high-dimensional, it is commonly represented in a low-dimensional embedding $\mathbf{e}_c = one\_hot(x_c)\mathbf{V}_c$ via embedding matrix $\mathbf{V}_c$.

## 3 FEATURE INTERACTIONS IN BLACK-BOX MODELS

We start by explaining how to obtain a data-instance level (local) interpretation of feature interactions by utilizing interaction detection on feature perturbations.

### 3.1 Feature Perturbation and Inference

Given a data instance $\mathbf{x} \in \mathbb{R}^p$, LIME proposed to perturb the data instance by sampling a separate binary representation $\tilde{\mathbf{x}} \in \{0, 1\}^d$ of the same data instance. Let $\xi : \{0, 1\}^d \to \mathbb{R}^p$ be the map from the binary representation to the perturbed data instance. Starting from a binary vector of all ones that map to the original features values in the data instance, LIME uniformly samples the number of random features to switch to $0$ or the "off" state. In the data instance, "off" could correspond to a $0$ embedding vector for categorical features or mean value over a batch for numerical features. It is possible for $d < p$ by grouping features in the data instance to correspond to single binary features in $\tilde{\mathbf{x}}$. An important step is getting black-box predictions of the perturbed data instances to create a dataset with binary inputs and prediction targets: $\mathcal{D} = \{(\tilde{\mathbf{x}}_i, y_i) \mid y_i = f(\xi(\tilde{\mathbf{x}}_i)), \tilde{\mathbf{x}}_i \in \{0, 1\}^d\}$. Though we use LIME's approach, the next section is agnostic to the instance perturbation method.

### 3.2 Feature Interaction Detection

Feature interaction detection is concerned with identifying feature interactions in a dataset (Bien et al., 2013; Purushotham et al., 2014; Lou et al., 2013; Friedman et al., 2008). Typically, proper interaction detection requires a pre-processing step to remove correlated features that adversely affect detection performance (Sorokina et al., 2008). As long as features in dataset $\mathcal{D}$ are generated in an uncorrelated fashion, e.g., through random sampling, we can directly use $\mathcal{D}$ to detect feature interactions from black-box model $f$ at data instance $\mathbf{x}$.

#### 3.2.1 Neural Interaction Detection

$f$ can be an arbitrary function and can generate highly nonlinear targets in $\mathcal{D}$, so we focus on detecting interactions that could have generic forms. In light of this, we leverage our method, Neural Interaction Detection (NID) (Tsang et al., 2017), which accurately and efficiently detects generic non-additive and arbitrary-order statistical feature interactions. NID detects these interactions by training a lasso-regularized multilayer perceptron (MLP) on a dataset, then identifying the features that have high-magnitude weights to common hidden units. NID is efficient by greedily testing the top-interaction candidates of every order at each of $h$ first-layer hidden units, enabling arbitrary-order interaction detection in $O(hd)$ tests within one MLP.

#### 3.2.2 Gradient-based Neural Interaction Detection

Besides the non-additive definition of statistical interaction, a gradient definition also exists based on mixed partial derivatives (Friedman et al., 2008), i.e., a function $F(\cdot)$ exhibits statistical interaction $\mathcal{I}$ among features $z_i$ indexed by $i_1, i_2, \ldots, i_{|\mathcal{I}|} \in \mathcal{I}$ if

$$E_{\mathbf{z}} \left[ \frac{\partial^{|\mathcal{I}|} F(\mathbf{z})}{\partial z_{i_1} \partial z_{i_2} \ldots \partial z_{i_{|\mathcal{I}|}}} \right]^2 > 0.$$

The advantage of this definition is that it allows exact interaction detection from model gradients (Ai & Norton, 2003); however, this definition contains a computationally expensive expectation, and typical neural networks with ReLU activation functions do not permit mixed partial derivatives. For the task of local interpretation, we only examine a single data instance $\mathbf{x}$, which avoids the expectation. We turn $F$ into an MLP $g(\cdot)$ with smooth, infinitely-differentiable activation functions such as softplus, which closely follows ReLU (Glorot et al., 2011). We then train the MLP with the same purpose as §3.2.1 to faithfully capture interactions in perturbation dataset $\mathcal{D}$. Given these conditions, we define an alternate gradient-based neural interaction detector (GradientNID) as:

$$\omega(\mathcal{I}) = \left( \frac{\partial^{|\mathcal{I}|} g(\tilde{\mathbf{x}})}{\partial \tilde{x}_{i_1} \partial \tilde{x}_{i_2} \ldots \partial \tilde{x}_{i_{|\mathcal{I}|}}} \right)^2,$$

where $\omega$ is the strength of the interaction $\mathcal{I}$, $\tilde{\mathbf{x}}$ is the representation of $\mathbf{x}$, and the MLP $g$ is trained on $\mathcal{D}$. While GradientNID exactly detects interactions from the explainer MLP, it needs to compute interaction strengths $\omega$ for feature combinations that grow exponentially in number as $|\mathcal{I}|$ increases. We recommend restricting GradientNID to low-order interactions.

---

**Algorithm 1** Global Interaction Detection in `GLIDER`

---

**Input:** dataset $\mathcal{B}$, recommender model $f_{rec}$
**Output:** $\mathcal{G} = \{(\mathcal{I}_i, c_i)\}$: global interactions $\mathcal{I}_i$ and their counts $c_i$ over the dataset
  1: $\mathcal{G} \leftarrow$ initialize occurrence dictionary for global interactions
  2: **for** each data sample $\mathbf{x}$ within dataset $\mathcal{B}$ **do**
  3:    $\mathcal{S} \leftarrow \text{MADEX}(f_{rec}, \mathbf{x})$
  4:    $\mathcal{G} \leftarrow$ increment the occurrence count of $\mathcal{I}_j \in \mathcal{S}, \forall j = 1, 2, \ldots, |\mathcal{S}|$
  5: sort $\mathcal{G}$ by most frequently occurring interactions
  6: [optional] prune subset interactions in $\mathcal{G}$ within a target number of interactions $K$

---

### 3.3 Scope

Based on §3.1 and §3.2, we define a function, $\text{MADEX}(f, \mathbf{x})$, that takes as inputs black-box $f$ and data instance $\mathbf{x}$, and outputs $\mathcal{S} = \{\mathcal{I}_i\}_{i=1}^k$, a set of top-$k$ detected feature interactions. `MADEX` stands for "Model-Agnostic Dependency Explainer".

In some cases, it is necessary to identify a $k$ threshold. Because of the importance of speed for local interpretations, we simply use a linear regression with additional multiplicative terms to approximate the gains given by interactions in $\mathcal{S}$, where $k$ starts at $0$ and is incremented until the linear model's predictions stop improving.

## 4 GLIDER: GLOBAL INTERACTION DETECTION AND ENCODING FOR RECOMMENDATION

We now discuss the different components of `GLIDER`: detecting global interactions in §4.1, then encoding these interactions in recommender systems in §4.2. Recommender systems are interesting because they have pervasive application in real-world systems, and their features are often very sparse. By sparse features, we mean features with many categories, e.g., millions of user IDs. The sparsity makes interaction detection challenging especially when applied directly on raw data because the one-hot encoding of sparse features creates an extremely large space of potential feature combinations (Fan et al., 2015).

### 4.1 Global Interaction Detection

In this section, we explain the first step of `GLIDER`. As defined in §3.3, `MADEX` takes as input a black-box model $f$ and data instance $\mathbf{x}$. In the context of this section, `MADEX` inputs a source recommender system $f_{rec}$ and data instance $\mathbf{x} = [x_1, x_2, \ldots, x_p]$. $x_i$ is the $i$-th feature field and is either a dense or sparse feature. $p$ is both the total number of feature fields and the number of perturbation variables ($p = d$). We define global interaction detection as repeatedly running `MADEX` over a batch of data instances, then counting the occurrences of the same detected interactions, shown in Algorithm 1. The occurrence counts are not only a useful way to rank global interaction detections, but also a sanity check to rule out the chance that the detected feature combinations are random selections.

One potential concern with Alg. 1 is that it could be slow depending on the speed of `MADEX`. In our experiments, the entire process took less than one hour when run in parallel over a batch of 1000 samples with $\sim 40$ features on a 32-CPU server with 2 GPUs. This algorithm only needs to be run once to obtain the summary of global interactions.

### 4.2 Truncated Feature Crosses

The global interaction $\mathcal{I}_i$, outputted by Alg. 1, is used to create a synthetic feature $x_{\mathcal{I}_i}$ for a target recommender system. The synthetic feature $x_{\mathcal{I}_i}$ is created by explicitly crossing sparse features indexed in $\mathcal{I}_i$. If interaction $\mathcal{I}_i$ involves dense features, we bucketize the dense features before crossing them. The synthetic feature is sometimes called a cross feature (Wang et al., 2017; Luo et al., 2019) or conjunction feature (Rosales et al., 2012; Chapelle et al., 2015).

In this context, a cross feature is an $n$-ary Cartesian product among $n$ sparse features. If we denote $\mathcal{X}_1, \mathcal{X}_2, \ldots, \mathcal{X}_n$ as the set of IDs for each respective feature $x_1, x_2, \ldots, x_n$, then their cross feature $x_{\{1,\ldots,n\}}$ takes on all possible values in

$$\mathcal{X}_1 \times \cdots \times \mathcal{X}_n = \{(x_1, \ldots, x_n) \mid x_i \in \mathcal{X}_i, \forall i = 1, \ldots, n\}$$

Accordingly, the cardinality of this cross feature is $|\mathcal{X}_1| \times \cdots \times |\mathcal{X}_n|$ and can be extremely large, yet many combinations of values in the cross feature are likely unseen in the training data. Therefore, we generate a truncated form of the cross feature with only seen combinations of values, $\mathbf{x}_{\mathcal{I}}^{(j)}$, where $j$ is a sample index in the training data, and $\mathbf{x}_{\mathcal{I}}^{(j)}$ is represented as a sparse ID in the cross feature $x_{\mathcal{I}}$. We further reduce the cardinality by requiring the same cross feature ID to occur more than $T$ times in a batch of samples, or set to a default ID otherwise. These truncation steps significantly reduce the embedding sizes of each cross feature while maintaining their representation power. Once cross features $\{x_{\mathcal{I}_i}\}_i$ are included in a target recommender system, it can be trained as per usual.

### 4.3 Model Distillation vs. Enhancement

There are dual perspectives of GLIDER: as a method for model distillation or model enhancement. If a strong source model is used to detect global interactions which are then encoded in more resource-constrained target models, then GLIDER adopts a teacher-student type distillation process. If interaction encoding augments the same model where the interactions were detected from, then GLIDER tries to enhance the model's ability to represent the interactions.

## 5 Related Works

**Interaction Interpretations**: A variety of methods exist to detect feature interactions learned in specific models but not black-box models. For example, RuleFit (Friedman et al., 2008), Additive Groves (Sorokina et al., 2008), and Tree-Shap (Lundberg et al., 2018) detect interactions specifically in trees; likewise PaD2 (Gevrey et al., 2006) and NID (Tsang et al., 2017) detect interactions in multilayer perceptrons. Some methods have attempted to interpret feature groups in black-box models, such as Anchors (Ribeiro et al., 2018), Agglomerative Contextual Decomposition (Singh et al., 2019), and Context-Aware methods (Singla et al., 2019); however, these methods were not intended to identify feature interactions.

**Explicit Interaction Representation**: There are increasingly methods for explicitly representing interactions in models. Cheng et al. (2016), Guo et al. (2017), Wang et al. (2017), and Lian et al. (2018) directly incorporate multiplicative cross terms in neural network architectures and Song et al. (2018) use attention as an interaction module, all of which are intended to improve the neural network's function approximation. This line of work found that predictive performance can improve with dedicated interaction modeling. Luo et al. (2019) followed up by proposing feature sets from data then explicitly encoding them via feature crossing, but this method's proposals are limited by beam search. Our work approaches this problem from a model interpretation standpoint.

**Black-Box Local vs. Global Interpretations**: Data-instance level local interpretation methods are more flexible at explaining general black-box models; however, global interpretations, which cover multiple data instances, have become increasingly desirable to summarize model behavior. Locally Interpretable Model-Agnostic Explanations (LIME) (Ribeiro et al., 2016) and Integrated Gradients (Sundararajan et al., 2017) are some of the most used methods to locally interpret any classifier and neural predictor respectively. There are some methods for global black-box interpretations, such as shuffle-based feature importance (Fisher et al., 2018), submodular pick (Ribeiro et al., 2016), and visual concept extraction (Kim et al., 2018). Our work offers a new tooling option.

## 6 Experiments

### 6.1 Setup

In our experiments, we study interaction interpretation and encoding on real-world data. The hyperparameters in MADEX are as follows. For all experiments, our perturbation datasets $\mathcal{D}$ contain 5000 training samples and 500 samples for each validation and testing. Our usage of NID or Gradient-NID as the interaction detector (§3.2) depends on the experimental setting. For all experiments that only examine single data instances, we use GradientNID for its exactness and pairwise interaction

detection; otherwise, we use NID for its higher-order interaction detection. The MLPs for NID and GradientNID have architectures of 256-128-64 first-to-last hidden layer sizes, and they are trained with learning rate of $1e-2$, batchsize of $100$, and the ADAM optimizer. NID uses ReLU activations and an $\ell_1$ regularization of $\lambda_1 = 1e-4$, whereas GradientNID uses softplus activations and a structural regularizer as MLP+linear regression, which we found offers strong test performance. In general, models are trained with early stopping on validation sets.

For LIME perturbations, we need to establish what a binary $0$ maps to via $\xi$ in the raw data instance (§3.1). In domains involving embeddings, i.e., sparse features and word embeddings, the $0$ ("off") state is the zeroed embedding vector. For dense features, it is the mean feature value over a batch; for images, the mean superpixel RGB of the image. For our DNA experiment, we use a random nucleotide other than the original one. These settings correspond to what is used in literature (Ribeiro et al., 2016; 2018). In our graph experiment, the nodes within the neighborhood of a test node are perturbed, where each node is zeroed during perturbation.

## 6.2 EXPERIMENTS ON CTR RECOMMENDATION

In this section, we provide experiments with GLIDER on models trained for click-through-rate (CTR) prediction. The recommender models we study include commonly reported baselines, which all use neural networks: Wide&Deep (Cheng et al., 2016), DeepFM (Guo et al., 2017), Deep&Cross (Wang et al., 2017), xDeepFM (Lian et al., 2018), and AutoInt (Song et al., 2018).

Table 1: CTR dataset statistics

| Dataset | # Samples | # Features | Total # Sparse IDs |
|---------|-----------|-----------|--------------------|
| Criteo | $45,840,617$ | $39$ | $998,960$ |
| Avazu | $40,428,967$ | $23$ | $1,544,428$ |

AutoInt is the reported state-of-the-art in academic literature, so we use the model settings and data splits provided by AutoInt's official public repository[1]. For all other recommender models, we use public implementations[2] with the same original architectures reported in literature, set all embedding sizes to 16, and tune the learning rate and optimizer to reach or surpass the test logloss reported by the AutoInt paper (on AutoInt's data splits). From tuning, we use the Adagrad optimizer (Duchi et al., 2011) with learning rate of $0.01$. All models use early stopping on validation sets.

The datasets we use are benchmark CTR datasets with the largest number of features: Criteo[3] and Avazu[4], whose data statistics are shown in Table 1. Criteo and Avazu both contain $40+$ millions of user records on clicking ads, with Criteo being the primary benchmark in CTR research (Cheng et al., 2016; Guo et al., 2017; Wang et al., 2017; Lian et al., 2018; Song et al., 2018; Luo et al., 2019).

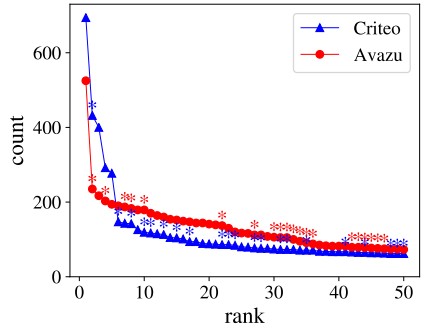

### 6.2.1 GLOBAL INTERACTION DETECTION

For each dataset, we train a source AutoInt model, $f_{rec}$, then run global interaction detection via Algorithm 1 on a batch of $1000$ samples from the validation set. A full global detection experiment finishes in less than one hour when run in parallel on either Criteo or Avazu datasets in a 32-CPU Intel Xeon E5-2640 v2 @ 2.00GHz server with 2 Nvidia 1080 Ti GPUs. The detection results across datasets are shown in Figure 2 as plots of detection counts versus rank. Because the Avazu dataset contains non-anonymized features, we directly show its top-10 detected global interactions in Table 2a.

Figure 2: Occurrence counts (Total: 1000) vs. rank of detected interactions from AutoInt on Criteo and Avazu datasets. * indicates a higher-order interaction (details in Appendix G).

---

[1] https://github.com/shichence/AutoInt
[2] https://github.com/shenweichen/DeepCTR
[3] https://www.kaggle.com/c/criteo-display-ad-challenge
[4] https://www.kaggle.com/c/avazu-ctr-prediction

Table 2: **Understanding feature interactions**: top global feature interactions for (a) an ad targeting system via Algorithm 1 and (b) a text sentiment analyzer via §6.3.2 (later). The tables are juxtaposed to assist in understanding feature interactions, i.e., nuanced changes among interacting variables lead to significant changes in prediction probabilities. The prediction outcomes are ad-clicks by users for (a) and text sentiment for (b).

| (a) Explanation of an ad targeting system | | (b) Explanation of a sentiment analyzer | |
|---|---|---|---|
| Count (Total:1000) | Interaction | Count (Total:40) | Interaction (ordered) |
| 525 | {device_ip, hour} | 36 | never, fails |
| 235 | {device_id, device_ip, hour} | 30 | suspend, disbelief |
| 217 | {device_id, app_id} | 30 | too, bad |
| 203 | {device_ip, device_model, hour} | 29 | very, funny |
| 194 | {site_id, site_domain} | 29 | neither, nor |
| 190 | {site_id, hour} | 28 | not, miss |
| 187 | {device_ip, site_id, hour} | 27 | recent, memory |
| 183 | {site_id, site_domain, hour} | 27 | not, good |
| 179 | {device_id, hour} | 26 | no, denying |
| 179 | {device_id, device_ip, device_model, hour} | 25 | not, bad |

From Figure 2, we see that the same interactions are detected very frequently across data instances, and many of the interactions are higher-order interactions. The interaction counts are very significant. For example, any top-1 occurrence count $> 25$ is significant for the Criteo dataset ($p < 0.05$), and likewise $> 71$ for the Avazu dataset, assuming a conservative search space of only up to 3-way interactions ($|\mathcal{I}| \leq 3$). Our top-1 occurrence counts are 691 ($\gg 25$) for Criteo and 525 ($\gg 71$) for Avazu.

In Table 2a, the top-interactions are explainable. For example, the interaction between "device_ip" and "hour" (in UTC time) makes sense because users - here identified by IP addresses - have ad-click behaviors dependent on their time zones. This is a general theme with many of the top-interactions[5]. As another example, the interaction between "device_id" and "app_id" makes sense because ads are targeted to users based on the app they're in.

### 6.2.2 Interaction Encoding

Based on our results from the previous section (§6.2.1), we turn our attention to explicitly encoding the detected global interactions in target baseline models via truncated feature crosses (detailed in §4.2). In order to generate valid cross feature IDs, we bucketize dense features into a maximum of 100 bins before crossing them and require that final cross feature IDs occur more than $T = 100$ times over a training batch of one million samples.

We take AutoInt's top-$K$ global interactions on each dataset from §6.2.1 with subset interactions excluded (Algorithm 1, line 6) and encode the interactions in each baseline model including AutoInt itself. $K$ is tuned on valiation sets, and model hyperparameters are the same between a baseline and one with encoded interactions. We set $K = 40$ for Criteo and $K = 10$ for Avazu.

In Table 3, we found that GLIDER often obtains significant gains in performance based on standard deviation, and GLIDER often reaches or exceeds a desired 0.001 improvement for the Criteo dataset (Cheng et al., 2016; Guo et al., 2017; Wang et al., 2017; Song et al., 2018). The improvements are especially visible with DeepFM on Criteo. We show how this model's test performance varies with different $K$ in Figure 3. All performance gains are obtained at limited cost of extra model parameters (Table 4) thanks to the truncations applied to our cross features. To avoid extra parameters entirely, we recommend feature selection on the new and existing features.

One one hand, the evidence that AutoInt's detected interactions can improve other baselines' performance suggests the viability of interaction distillation. On the other hand, evidence that AutoInt's performance on Criteo can improve using its own detected interactions suggests that AutoInt may benefit from learning interactions more explicitly. In either model distillation or enhancement set-

---

[5]"device_ip" and "device_id" identify different sets of users (https://www.csie.ntu.edu.tw/~r01922136/slides/kaggle-avazu.pdf)

Table 3: **Test prediction performance** by encoding top-$K$ global interactions in baseline recommender systems on the Criteo and Avazu datasets (5 trials). $K$ are 40 and 10 for Criteo and Avazu respectively. "+ GLIDER" means the inclusion of detected global interactions to corresponding baselines. The "Setting" column is labeled relative to the source of detected interactions: AutoInt. * scores by Song et al. (2018).

| Setting | Model | Criteo | | Avazu | |
|---|---|---|---|---|---|
| | | AUC | logloss | AUC | logloss |
| Distillation | Wide&Deep | $0.8069 \pm 5e-4$ | $0.4446 \pm 4e-4$ | $0.7794 \pm 3e-4$ | $0.3804 \pm 2e-4$ |
| | + GLIDER | $\mathbf{0.8080 \pm 3e-4}$ | $\mathbf{0.4436 \pm 3e-4}$ | $0.7795 \pm 1e-4$ | $\mathbf{0.3802 \pm 9e-5}$ |
| | DeepFM | $0.8079 \pm 3e-4$ | $0.4436 \pm 2e-4$ | $0.7792 \pm 3e-4$ | $0.3804 \pm 9e-5$ |
| | + GLIDER | $\mathbf{0.8097 \pm 2e-4}$ | $\mathbf{0.4420 \pm 2e-4}$ | $\mathbf{0.7795 \pm 2e-4}$ | $\mathbf{0.3802 \pm 2e-4}$ |
| | Deep&Cross | $0.8076 \pm 2e-4$ | $0.4438 \pm 2e-4$ | $0.7791 \pm 2e-4$ | $0.3805 \pm 1e-4$ |
| | + GLIDER | $\mathbf{0.8086 \pm 3e-4}$ | $\mathbf{0.4428 \pm 2e-4}$ | $0.7792 \pm 2e-4$ | $\mathbf{0.3803 \pm 9e-5}$ |
| | xDeepFM | $0.8084 \pm 2e-4$ | $0.4433 \pm 2e-4$ | $0.7785 \pm 3e-4$ | $0.3808 \pm 2e-4$ |
| | + GLIDER | $\mathbf{0.8097 \pm 3e-4}$ | $\mathbf{0.4421 \pm 3e-4}$ | $0.7787 \pm 4e-4$ | $\mathbf{0.3806 \pm 1e-4}$ |
| Enhancement | AutoInt * | $0.8083$ | $0.4434$ | $0.7774$ | $0.3811$ |
| | + GLIDER | $\mathbf{0.8090 \pm 2e-4}$ | $\mathbf{0.4426 \pm 2e-4}$ | $0.7773 \pm 1e-4$ | $0.3811 \pm 5e-5$ |

Table 4: # parameters of the models in Table 3. M denotes million.

| Model | Criteo | Avazu |
|---|---|---|
| Wide&Deep | 18.1M | 27.3M |
| + GLIDER | 19.3M (+6.8%) | 27.6M (+1.0%) |
| DeepFM | 17.5M | 26.7M |
| + GLIDER | 18.3M (+4.8%) | 26.9M (+0.6%) |
| Deep&Cross | 17.5M | 26.1M |
| + GLIDER | 18.7M (+6.9%) | 26.4M (+1.0%) |
| xDeepFM | 18.5M | 27.6M |
| + GLIDER | 21.7M (+17.2%) | 28.3M (+2.5%) |
| AutoInt | 16.4M | 25.1M |
| + GLIDER | 17.3M (+5.1%) | 25.2M (+0.6%) |

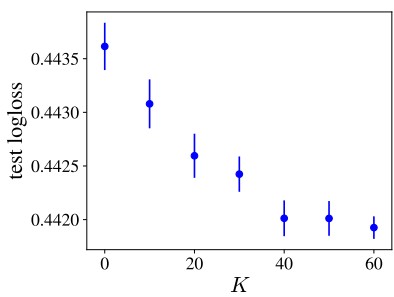

Figure 3: Test logloss vs. $K$ of DeepFM on the Criteo dataset (5 trials).

tings, we found that GLIDER performs especially well on industry production models trained on large private datasets with thousands of features.

## 6.3 INTERPRETATIONS ON OTHER DOMAINS

Since the proposed interaction interpretations are not entirely limited to recommender systems, we demonstrate interpretations on more general black-box models. Specifically, we experiment with the function MADEX($\cdot$) defined in §3.3, which inputs a black-box $f$, data-instance $\mathbf{x}$, and outputs a set of top-$k$ interactions. The models we use are trained on very different tasks, i.e., ResNet152: an image classifier pretrained on ImageNet '14 (Russakovsky et al., 2015; He et al., 2016), Sentiment-LSTM: a 2-layer bi-directional long short-term memory network (LSTM) trained on the Stanford Sentiment Treebank (SST) (Socher et al., 2013; Tai et al., 2015), DNA-CNN: a 2-layer 1D convolutional neural network (CNN) trained on MYC-DNA binding data[6] (Mordelet et al., 2013; Yang et al., 2013; Alipanahi et al., 2015; Zeng et al., 2016; Wang et al., 2018; Barrett et al., 2012), and GCN: a 3-layer Graph Convolutional Network trained on the Cora dataset (Kipf & Welling, 2016; Sen et al., 2008). In order to make informative comparisons to the linear LIME baseline, we use LIME's sample weighting strategy and kernel size (0.25) in this section. We first provide quantitative validation for the detected interactions of all four models in §6.3.1, followed by qualitative results for ResNet152, Sentiment-LSTM, and DNA-CNN in §6.3.2.

### 6.3.1 QUANTITATIVE

To quantitatively validate our interaction interpretations of general black-box models, we measure the local explanation fidelity of the interactions via prediction performance. As suggested in §3.3 and

---

[6]https://www.ncbi.nlm.nih.gov/geo/query/acc.cgi?acc=GSE47026

Table 5: Prediction performance (mean-squared error; lower is better) with ($k > 0$) and without ($k = 0$) interactions for random data instances in the test sets of respective black-box models. $k = L$ corresponds to the interaction at a rank threshold. $2 \leq k < L$ are excluded because not all instances have 2 or more interactions. Only results with detected interactions are shown. At least $94\%$ ($\geq 188$) of the data instances had interactions across 5 trials for each model and score statistic.

| | $k$ | DNA-CNN | Sentiment-LSTM | ResNet152 | GCN |
|---|---|---|---|---|---|
| linear LIME | 0 | $10e{-}3 \pm 1e{-}3$ | $8.0e{-}2 \pm 6e{-}3$ | $1.9 \pm 0.1$ | $7.1e3 \pm 7e2$ |
| MADEX (ours) | 1 | $8e{-}3 \pm 2e{-}3$ | $3.8e{-}2 \pm 6e{-}3$ | $1.7 \pm 0.1$ | $5.7e3 \pm 7e2$ |
| MADEX (ours) | $L$ | $5.0e{-}3 \pm 8e{-}4$ | $0.4e{-}2 \pm 3e{-}3$ | $0.9 \pm 0.2$ | $2e3 \pm 1e3$ |

§4.2, encoding feature interactions is a way to increase a model's function representation, but this also means that prediction performance gains over simpler first-order models (e.g., linear regression) is a way to test the significance of the detected interactions. In this section, we use neural network function approximators for each top-interaction from the ranking $\{\mathcal{I}_i\}$ given by MADEX's interaction detector (in this case NID). Similar to the $k$-thresholding description in §3.3, we start at $k = 0$, which is a linear regression, then increment $k$ with added MLPs for each $\mathcal{I}_i$ among $\{\mathcal{I}_i\}_{i=1}^k$ until validation performance stops improving, denoted at $k = L$. The MLPs all have architectures of 64-32-16 first-to-last hidden layer sizes and use the binary perturbation dataset $\mathcal{D}$ (from §3.1).

Test prediction performances are shown in Table 5 for $k \in \{0, 1, L\}$. The average number of features of $\mathcal{D}$ among the black-box models ranges from 18 to 112. Our quantitative validation shows that adding feature interactions for DNA-CNN, Sentiment-LSTM, and ResNet152, and adding node interactions for GCN result in significant performance gains when averaged over 40 randomly selected data instances in the test set.

### 6.3.2 QUALITATIVE

For our qualitative analysis, we provide interaction interpretations via MADEX($\cdot$) of ResNet152, Sentiment-LSTM, and DNA-CNN on test samples. The interpretations are given by $\mathcal{S} = \{\mathcal{I}_i\}_{i=1}^k$, a set of $k$ detected interactions, which are shown in Figure 4 for ResNet152 and Sentiment-LSTM. For reference, we also show the top "main effects" by LIME's original linear regression, which select the top-5 features that attribute towards the predicted class[7].

In Figure 4a, the "interaction" columns show selected features from MADEX's interactions between Quickshift superpixels (Vedaldi & Soatto, 2008; Ribeiro et al., 2016). To reduce the number of interactions per image, we merged interactions that have overlap coefficient $\geq$ 0.5 (Vijaymeena & Kavitha, 2016). From

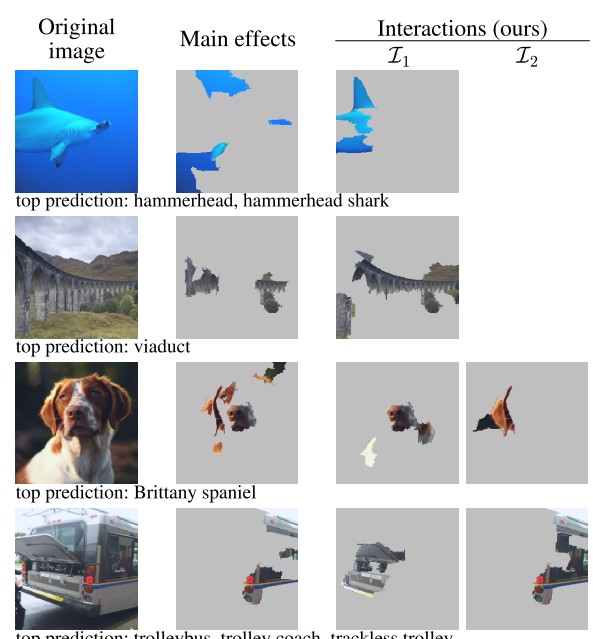

(a) ResNet152 interpretations

| Original sentence | Prediction | Main effects | Interactions (ours) | |
|---|---|---|---|---|
| | | | $\mathcal{I}_1$ | $\mathcal{I}_2$ |
| It never fails to engage us. | pos. | never, us | never, fails | |
| The movie makes absolutely no sense. | neg. | no, sense | absolutely, no | no, sense |
| The central story lacks punch. | neg. | lacks | story, lacks | lacks, punch |

(b) Sentiment-LSTM interpretations

Figure 4: Qualitative examples (more in Appendix D & E)

---

[7]Based on official code: `https://github.com/marcotcr/lime`

the figure, we see that the interactions form a single region or multiple regions of the image. They also tend to be complementary to LIME's main effects and are sometimes more informative. For example, the interpretations of the "shark" classification show that interaction detection finds the shark fin whereas main effects do not. Interpretations of Sentiment-LSTM are shown in Figure 4b, excluding common stop words (Appendix C). We again see the value of MADEX's interactions, which show salient combinations of words, such as "never, fails", "absolutely, no", and "lacks, punch".

In our experiments on DNA-CNN, we consistently detected the interaction between "CACGTG" nucleotides, which form a canonical DNA sequence (Staiger et al., 1989). The interaction was detected 97.3% out of 187 CACGTG appearances in the test set.

In order to run consistency experiments now on Sentiment-LSTM, word interactions need to be detected consistently across different sentences, which naïvely would require an exorbitant amount of sentences. Instead, we initially collect interaction candidates by running MADEX over all sentences in the SST test set, then select the word interactions that appear multiple times. We assume that word interactions are ordered but not necessarily adjacent or positionally bound, e.g., (not, good) $\neq$ (good, not), but their exact positions don't matter. We use the larger IMDB dataset (Maas et al., 2011) to collect different sets of sentences that contain the same ordered words as each interaction candidate (but the sentences are otherwise random). The ranked detection counts of the target interactions on their individual sets of sentences are shown in Table 2b. The average sentence length is 33 words, and interaction occurrences are separated by 2 words on average.

## 7 CONCLUSION

We proposed a way to interpret feature interactions in general prediction models, and we proposed GLIDER to detect and encode these interactions in black-box recommender systems. In our experiments on recommendation, we found that our detected global interactions are explainable and that explicitly encoding them can improve predictions. We further validated our interaction interpretations on image, text, graph, and dna models. We hope the interpretations encourage investigation into the complex behaviors of prediction models, especially models with large societal impact. Some opportunities for future work are generating correct attributions for interaction interpretations, preventing false-positive interactions from out-of-distribution feature perturbations, and performing interaction distillation from multiple models rather than just one.

ACKNOWLEDGMENTS

We would like to sincerely thank everyone who has provided their generous feedback for this work. Thank you Youbang Sun, Dongxu Ren, and Beibei Xin for offering early-stage brainstorming and prolonged discussions. Thank you Yuping Luo for providing advice on theoretical analysis of model interpretation. Thank you Rich Caruana for your support and insight. Thank you Artem Volkhin, Levent Ertoz, Ellie Wen, Long Jin, Dario Garcia, and the rest of the Facebook personalization team for your feedback on the paper content. Last but not least, thank you anonymous reviewers for your thorough comments and suggestions. This work was supported by National Science Foundation Awards IIS-1254206 and IIS-1539608, granted to co-author Yan Liu in her academic role at the University of Southern California.

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

## A  EFFECT OF EXTRA PARAMETERS BY INTERACTION ENCODINGS VS. ENLARGED EMBEDDINGS

In this section, we study whether increasing embedding size can obtain similar prediction performance gains as explicitly encoding interactions via GLIDER. We increase the embedding dimension sizes of every sparse feature in baseline recommender models to match the total number of model parameters of baseline + GLIDER as close as possible. The embedding sizes we used to obtain similar parameter counts are shown in Table 6. For the Avazu dataset, all of the embedding sizes remain unchanged because they were already the target size. The corresponding prediction performances of all models are shown in Table 7. We observed that directly increasing embedding size / parameter counts generally did not give the same level of performance gains that GLIDER provided.

Table 6: Comparison of # model parameters between baseline models with enlarged embeddings and original baselines + GLIDER (from Tables 3 and 4). The models with enlarged embeddings are denoted by the asterick (*). The embedding dimension of sparse features is denoted by "emb. size". Percent differences are relative to baseline* models. M denotes million, and the ditto mark (") means no change in the above line.

| Model | Criteo | | Avazu | |
|---|---|---|---|---|
| | emb. size | # params | emb. size | # params |
| Wide&Deep* | 17 | 19.1M | 16 | 27.3M |
| Wide&Deep | 16 | 18.1M | 16 | " |
| + GLIDER | 16 | 19.3M (+1.1%) | 16 | 27.6M (+1.0%) |
| DeepFM* | 17 | 18.5M | 16 | 26.7M |
| DeepFM | 16 | 17.5M | 16 | " |
| + GLIDER | 16 | 18.3M (−0.9%) | 16 | 26.9M (+0.6%) |
| Deep&Cross* | 17 | 18.5M | 16 | 26.1M |
| Deep&Cross | 16 | 17.5M | 16 | " |
| + GLIDER | 16 | 18.7M (+1.0%) | 16 | 26.4M (+1.0%) |
| xDeepFM* | 19 | 21.5M | 16 | 27.6M |
| xDeepFM | 16 | 18.5M | 16 | " |
| + GLIDER | 16 | 21.7M (+0.7%) | 16 | 28.3M (+2.5%) |
| AutoInt* | 17 | 17.4M | 16 | 25.1M |
| AutoInt | 16 | 16.4M | 16 | " |
| + GLIDER | 16 | 17.3M (−1.0%) | 16 | 25.2M (+0.6%) |

Table 7: Test prediction performance corresponding to the models shown in Table 6

| Model | Criteo | | Avazu | |
|---|---|---|---|---|
| | AUC | logloss | AUC | logloss |
| Wide&Deep* | $0.8072 \pm 3\text{e}{-4}$ | $0.4443 \pm 2\text{e}{-4}$ | $0.7794 \pm 3\text{e}{-4}$ | $0.3804 \pm 2\text{e}{-4}$ |
| Wide&Deep | $0.8069 \pm 5\text{e}{-4}$ | $0.4446 \pm 4\text{e}{-4}$ | " | " |
| + GLIDER | $\mathbf{0.8080 \pm 3\text{e}{-4}}$ | $\mathbf{0.4436 \pm 3\text{e}{-4}}$ | $0.7795 \pm 1\text{e}{-4}$ | $\mathbf{0.3802 \pm 9\text{e}{-5}}$ |
| DeepFM* | $0.8080 \pm 4\text{e}{-4}$ | $0.4435 \pm 4\text{e}{-4}$ | $0.7792 \pm 3\text{e}{-4}$ | $0.3804 \pm 9\text{e}{-5}$ |
| DeepFM | $0.8079 \pm 3\text{e}{-4}$ | $0.4436 \pm 2\text{e}{-4}$ | " | " |
| + GLIDER | $\mathbf{0.8097 \pm 2\text{e}{-4}}$ | $\mathbf{0.4420 \pm 2\text{e}{-4}}$ | $\mathbf{0.7795 \pm 2\text{e}{-4}}$ | $\mathbf{0.3802 \pm 2\text{e}{-4}}$ |
| Deep&Cross* | $0.8081 \pm 2\text{e}{-4}$ | $0.4434 \pm 2\text{e}{-4}$ | $0.7791 \pm 2\text{e}{-4}$ | $0.3805 \pm 1\text{e}{-4}$ |
| Deep&Cross | $0.8076 \pm 2\text{e}{-4}$ | $0.4438 \pm 2\text{e}{-4}$ | " | " |
| + GLIDER | $\mathbf{0.8086 \pm 3\text{e}{-4}}$ | $\mathbf{0.4428 \pm 2\text{e}{-4}}$ | $0.7792 \pm 2\text{e}{-4}$ | $\mathbf{0.3803 \pm 9\text{e}{-5}}$ |
| xDeepFM* | $0.8088 \pm 1\text{e}{-4}$ | $0.4429 \pm 1\text{e}{-4}$ | $0.7785 \pm 3\text{e}{-4}$ | $0.3808 \pm 2\text{e}{-4}$ |
| xDeepFM | $0.8084 \pm 2\text{e}{-4}$ | $0.4433 \pm 2\text{e}{-4}$ | " | " |
| + GLIDER | $\mathbf{0.8097 \pm 3\text{e}{-4}}$ | $\mathbf{0.4421 \pm 3\text{e}{-4}}$ | $0.7787 \pm 4\text{e}{-4}$ | $\mathbf{0.3806 \pm 1\text{e}{-4}}$ |
| AutoInt* | $0.8087 \pm 2\text{e}{-4}$ | $0.4431 \pm 1\text{e}{-4}$ | $\mathbf{0.7774 \pm 1\text{e}{-4}}$ | $0.3811 \pm 8\text{e}{-5}$ |
| AutoInt | $0.8083$ | $0.4434$ | " | " |
| + GLIDER | $\mathbf{0.8090 \pm 2\text{e}{-4}}$ | $\mathbf{0.4426 \pm 2\text{e}{-4}}$ | $0.7773 \pm 1\text{e}{-4}$ | $0.3811 \pm 5\text{e}{-5}$ |

## B  EFFECT OF DENSE FEATURE BUCKETIZATION

We examine the effect of dense feature bucketization on cross feature parameter efficiency for the Criteo dataset, which contains 13 dense features. Figure 5 shows the effects of varying the number of dense buckets on the embedding sizes of the cross features involving dense features. Both the effects on the average and individual embedding size are shown. 14 out of 40 of the cross features involved a dense feature. Different cross features show different parameter patterns as the number of buckets increases (Figure 5b). One one hand, the parameter count sometimes increases then asymptotes. Our requirement that a valid cross feature ID occurs more than $T$ times (§4.2) restricts the growth in parameters. On the other hand, the parameter count sometimes decreases, which happens when the dense bucket size becomes too small to satisfy the $T$ occurrence restriction. In all cases, the parameter counts are kept limited, which is important for overall parameter efficiency.

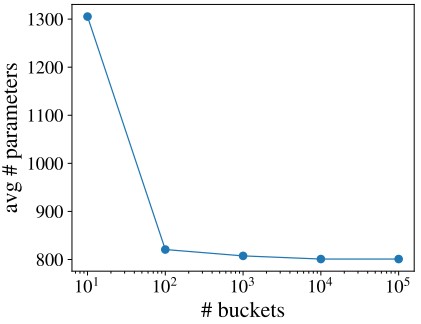

(a) effect on avg. cross feature embedding size

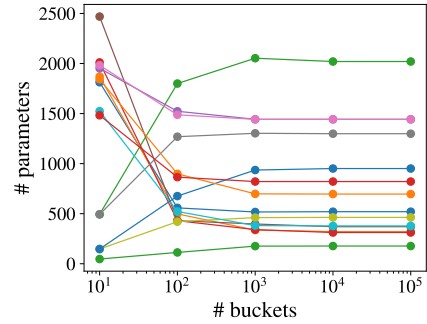

(b) effect on each cross feature embedding size

Figure 5: The effects of varying the number of buckets on (a) on the average embedding size of cross features involving dense features and (b) the individual embedding sizes of the same cross features.

## C    STOP WORDS

For all qualitative interpretations on text (in §6.3.2 and Appendix D), we preprocessed sentences to remove stop words. We use the same stop words suggested by Manning et al. (2008), i.e., {a, an, and, are, as, at, be, by, for, from, has, he, in, is, it, its, of, on, that, the, to, was, were, will, with}.

## D    QUALITATIVE RESULTS ON SENTIMENT-LSTM VS. BERT

In this section, we compare the word interactions discovered by MADEX on Sentiment-LSTM versus BERT. These models perform with accuracies of $87\%$ and $92\%$ respectively on the SST test set. We use a public pre-trained BERT, i.e., DistilBERT (Sanh et al., 2019), which is available online[8]. The interaction detector we use is GradientNID (§3.2.2), and sample weighting is disabled for this comparison. The top-2 interactions for each model are shown in Table 8 on random sentences from the SST test set.

Table 8: Top-ranked word interactions $\mathcal{I}_i$ from Sentiment-LSTM and BERT on randomly selected sentences in the SST test set.

| Original sentence | Sentiment-LSTM | | BERT | |
| --- | --- | --- | --- | --- |
| | $\mathcal{I}_1$ | $\mathcal{I}_2$ | $\mathcal{I}_1$ | $\mathcal{I}_2$ |
| An intelligent, earnest, intimate film that drops the ball only when it pauses for blunt exposition to make sure you're getting its metaphysical point. | intelligent, metaphysical | metaphysical, point | intelligent, earnest | drops, ball |
| It's not so much enjoyable to watch as it is enlightening to listen to new sides of a previous reality, and to visit with some of the people who were able to make an impact in the theater world. | not, enjoyable | not, so | not, much | not, enlightening |
| Uneasy mishmash of styles and genres. | uneasy, mishmash | mishmash, genres | uneasy, mishmash | uneasy, styles |
| You're better off staying home and watching the X-Files. | x, files | off, x | better, off | you, off |
| If this is the Danish idea of a good time, prospective tourists might want to consider a different destination – some jolly country embroiled in a bloody civil war, perhaps. | if, this | if, good | if, jolly | jolly, country |
| We can see the wheels turning, and we might resent it sometimes, but this is still a nice little picture, made by bright and friendly souls with a lot of good cheer. | resent, nice | we, resent | nice, good | nice, made |
| One of the greatest family-oriented, fantasy-adventure movies ever. | family, oriented | greatest, family | greatest, family | adventure, movies |
| It's so full of wrong choices that all you can do is shake your head in disbelief – and worry about what classic Oliver Parker intends to mangle next time. | so, wrong | full, wrong | so, wrong | so, full |
| Its mysteries are transparently obvious, and it's too slowly paced to be a thriller. | mysteries, transparently | paced, thriller | too, thriller | too, paced |
| This miserable excuse of a movie runs on empty, believing flatbush machismo will get it through. | miserable, runs | excuse, get | runs, empty | miserable, runs |

---

[8]https://github.com/huggingface/transformers

# E    ADDITIONAL QUALITATIVE RESULTS FOR RESNET152

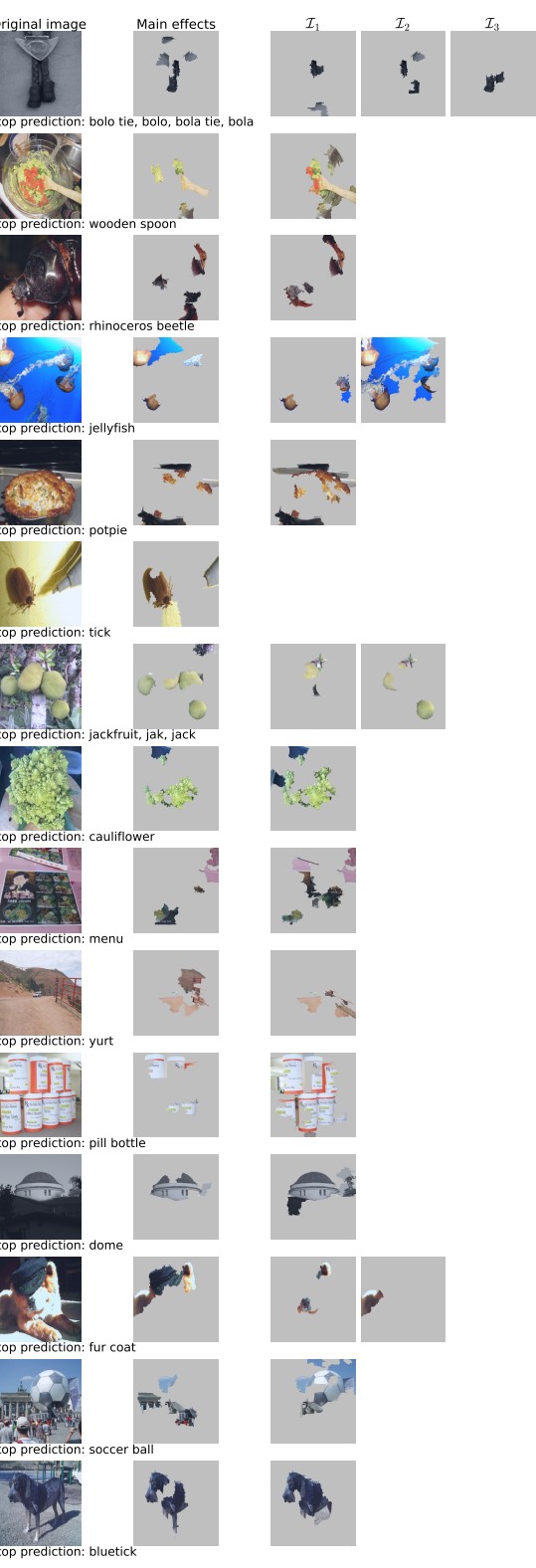

Figure 6: Additional qualitative results, following Figure 4a, on random test images in ImageNet. Interactions are denoted by $\mathcal{I}_i$ and are unordered. Overlapping interactions with overlap coefficient $\geq 0.5$ are merged to reduce $|\{\mathcal{I}_i\}|$ per test image.

## F  DETECTION PERFORMANCE OF MADEX VS. BASELINES

We compare the detection performances between MADEX and baselines on identifying feature interactions learned by complex models, i.e., XGBoost (Chen & Guestrin, 2016), Multilayer Perceptron (MLP), and Long Short-Term Memory Network (LSTM) (Hochreiter & Schmidhuber, 1997). The baselines are Tree-Shap: a method to identify interactions in tree-based models like XGBoost (Lundberg et al., 2018), MLP-ACD+: a modified version of ACD (Singh et al., 2019; Murdoch et al., 2018) to search all pairs of features in MLP to find the best interaction candidate, and LSTM-ACD+: the same as MLP-ACD+ but for LSTMs. All baselines are local interpretation methods. For MADEX, we sample continuous features from a truncated normal distribution $\mathcal{N}(\mathbf{x}, \sigma^2 \mathbf{I})$ centered at a specified data instance $\mathbf{x}$ and truncated at $\sigma$. Our MADEX experiments consist of two methods, NID and GradNID (shorthand for GradientNID).

We evaluate interaction detection performance by using synthetic data where ground truth interactions are known (Hooker, 2004; Sorokina et al., 2008). We generate 10e3 samples of synthetic data using functions $F_1 - F_4$ (Table 9) with continuous features uniformly distributed between $-1$ to $1$. Next, we train complex models (XGBoost, MLP, and LSTM) on this data. Lastly, we run MADEX and the baselines on 10 trials of 20 data instances at randomly sampled locations on the synthetic function domain. Between trials, the complex models are trained with different random initialization to test the stability of each interpretation method. Inter-

Table 9: Data generating functions with interactions

| | |
|---|---|
| $F_1(\mathbf{x}) =$ | $10x_1x_2 + \sum_{i=3}^{10} x_i$ |
| $F_2(\mathbf{x}) =$ | $x_1x_2 + \sum_{i=3}^{10} x_i$ |
| $F_3(\mathbf{x}) =$ | $\exp(|x_1 + x_2|) + \sum_{i=3}^{10} x_i$ |
| $F_4(\mathbf{x}) =$ | $10x_1x_2x_3 + \sum_{i=4}^{10} x_i$ |

action detection performance is computed by the average R-precision (Manning et al., 2008)[9] of interaction rankings across the sampled data instances.

Results are shown in Table 10. MADEX (NID and GradNID) performs well compared to the baselines. On the tree-based model, MADEX can compete with the tree-specific baseline Tree-Shap, which only detects pairwise interactions. On MLP and LSTM, MADEX performs significantly better than ACD+. The performance gain is especially large in the LSTM setting. Comparing NID and GradNID, NID tends to perform better in this experiment because it takes its entire sampling region into account whereas GradNID examines a single data instance.

Table 10: Detection Performance in R-Precision (higher the better). $\sigma = 0.6$ (max: 3.2). "Tree" is XGBoost. *Does not detect higher-order interactions. †Requires an exhaustive search of all feature combinations.

| | Tree | | | MLP | | | LSTM | | |
|---|---|---|---|---|---|---|---|---|---|
| | Tree-Shap | NID | GradNID | MLP-ACD+ | NID | GradNID | LSTM-ACD+ | NID | GradNID |
| $F_1(\mathbf{x})$ | $1 \pm 0$ | $1 \pm 0$ | $0.96 \pm 0.04$ | $0.63 \pm 0.08$ | $1 \pm 0$ | $1 \pm 0$ | $0.3 \pm 0.2$ | $1 \pm 0$ | $1 \pm 0$ |
| $F_2(\mathbf{x})$ | $1 \pm 0$ | $0.3 \pm 0.4$ | $0.6 \pm 0.4$ | $0.41 \pm 0.06$ | $1 \pm 0$ | $0.95 \pm 0.04$ | $0.01 \pm 0.02$ | $0.99 \pm 0.02$ | $0.95 \pm 0.04$ |
| $F_3(\mathbf{x})$ | $1 \pm 0$ | $1 \pm 0$ | $1 \pm 0$ | $0.3 \pm 0.2$ | $1 \pm 0$ | $1 \pm 0$ | $0.05 \pm 0.08$ | $1 \pm 0$ | $1 \pm 0$ |
| $F_4(\mathbf{x})$ | * | $1 \pm 0$ | † | † | $1 \pm 0$ | † | † | $1 \pm 0$ | † |

---

[9]R-precision is the percentage of the top-$R$ items in a ranking that are correct out of $R$, the number of correct items. $R = 1$ in these experiments.

## G  HIGHER-ORDER INTERACTIONS

This section shows how often different orders of higher-order interactions are identified by `GLIDER` / `MADEX`. Figure 7 plots the occurrence counts of global interactions detected in AutoInt for the Criteo and Avazu dataset, which correspond to the results in Figure 2. Here we show the occurrence counts of higher-order interactions, where the exact interaction cardinality is annotated besides each data point. 3-way interactions are the most common type, followed by 4-, then 5-way interactions.

Figure 8 plots histograms of interaction cardinalities for all interactions detected from ResNet152 and Sentiment-LSTM across 1000 random samples in their test sets. The average number of features are 66 and 18 for ResNet152 and Sentiment-LSTM respectively. Higher-order interactions are common in both models.

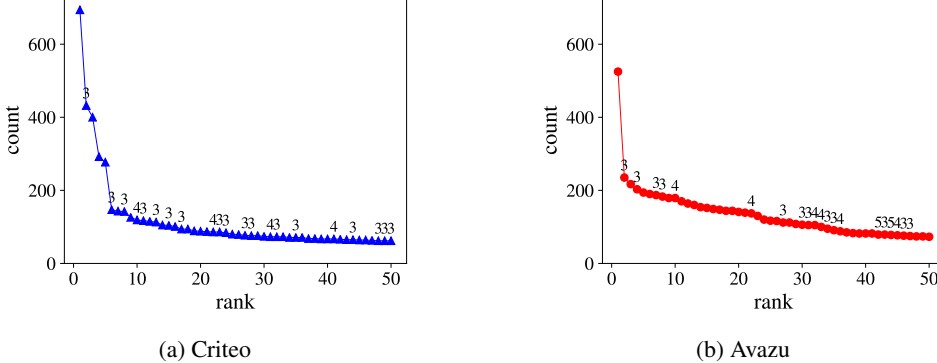

(a) Criteo  (b) Avazu

Figure 7: Occurrence counts (total: 1000) vs. rank of interactions detected from AutoInt on (a) Criteo and (b) Avazu datasets. Each higher-order interaction is annotated with its interaction cardinality.

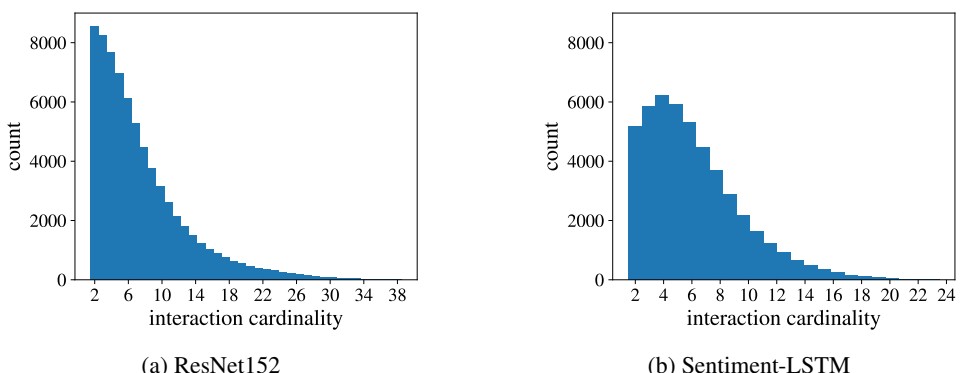

(a) ResNet152  (b) Sentiment-LSTM

Figure 8: Histograms of interaction sizes for interactions detected in (a) ResNet152 and (b) Sentiment-LSTM across 1000 random samples in respective test sets.

