# OpenReview forum: "Feature Interaction Interpretability: A Case for Explaining Ad-Recommendation Systems via Neural Interaction Detection"
_ICLR.cc/2020/Conference — Accept (Poster)_

### Official Review · AnonReviewer3 · 2019-10-24
**Official Blind Review #3**

**Rating:** 6

**Review:**

This paper proposed a model for extracting global feature interactions from the source model which was later being encoded in the target model to enhance its prediction performance.

Strong points:
1. The paper laid out the necessary background knowledge very clear even for the audiences outside this area.

2. The paper performed reasonable amount of experiments to compare the proposed model with various baseline models with various data sets, which are strong enough to support the claims made in this paper.

Comments:
1. The theoretical innovation of this paper is trivial. The local detection model for feature interactions, i.e. MADEX is simply employing the previous work, LIME and NID. Its global extension, i.e. GLIDER and the proposed encoding method are straightforward.

2. The descriptions on the feature dimensions of the original data and the generated binary representation data x’ are rather confusing and inconsistent throughout the paper. If I understand it correctly, the data sample from the original feature space x \in R^p, and a  binary representation x’ \in R^d, where d <= p. However, when d first appears in the first paragraph of section 3, it is defined as the dimension of the original feature space and later in that paragraph the dimension became p in “f(.): R^p -> R”. There is no clarification on the differences between d and p till section 4.1 where it states “x \in R^p ”. However, the dimension of the data instance from the original feature space changed to d again in section 4.2 where it states “x =[x_1, x_2, …, x_d]”.

3. “What’s more, the same approach to interpreting interactions can provide new insights into domains even beyond recommendation.” should be “What’s more, the same approach to interpret interactions can provide new insights into domains even beyond the recommendation.”

4. “An interaction, I, is a subset of all input features...”, but according to the following sections, I is the indices of a feature subset instead of the features themselves.

5. “...by requiring the same cross feature ID to occur more that T times in a batch of samples,...”, ‘that’ should be ‘than’.

Overall, although there is no significant theoretical innovation, it is a decent application paper.


**Experience Assessment:**

I do not know much about this area.

**Review Assessment: Checking Correctness Of Derivations And Theory:**

I carefully checked the derivations and theory.

**Review Assessment: Checking Correctness Of Experiments:**

I carefully checked the experiments.

**Review Assessment: Thoroughness In Paper Reading:**

I read the paper at least twice and used my best judgement in assessing the paper.

---

> ### Author Response · Authors · 2019-11-13
> **Author Response**
>
> Thank you for your supportive comments. We clarified the meanings of p, d and the I symbol, and we fixed the typos.
>
> We are very grateful that you recognize value in this work. Interpretability research has made progress in both theoretical and applied topics. Indeed this work is in the applied category. One of the focuses on the applied research end has been interpreting black-box models [1,2,3,4], which gave us strong motivation to study this direction. A glaring gap in this direction was the lack of discourse on interpreting feature interactions, which are important as we showed in experiments. We could not stop there, so we investigated how to leverage these interpretations: a type of problem that very few interpretability works have studied, especially for improving predictions automatically.
>
>
> [1] Marco Ribeiro, Sameer Singh, Carlos Guestrin. Why should I trust you?: Explaining the predictions of any classifier. In Proceedings of KDD, 2016.
> [2] Been Kim, Martin Wattenberg, Justin Gilmer, Carrie Cai, James Wexler, Fernanda Viegas, Rory Sayres. Interpretability beyond feature attribution: Quantitative testing with concept activation vectors (tcav). In Proceedings of ICML, 2018.
> [3] Pang Wei Koh, Percy Liang. Understanding black-box predictions via influence functions. In Proceedings of ICML, 2017.
> [4] Christoph Molnar. Interpretable machine learning. A guide for making black box models explainable. 2019.

---

### Official Review · AnonReviewer1 · 2019-10-24
**Official Blind Review #1**

**Rating:** 6

**Review:**

The paper proposes a method to detect which features in the input of recommender systems are interacted each other, i.e., combining them behaves useful information, and examines to feed extracted interactions directly into the recommender systems to measure effects on actual recommendation.

The interaction detector consists of 1. perturbing the input vectors,  2. training NNs to utilize its internal non-linear representation as a signal of interaction, and 3. aggregating detected interactions over training set. 1. and 2. are consisting of known methods so that the proposed method is an application of them. For 3. authors introduced a simple heuristic (Algorithm 1).

As long as I heard how the NID work to detect interaction from this paper, it is sensitive not only the true interaction between features but also set of features holding similar signals (e.g., if a hidden unit behaves as a feature A, it may be natural to aggregate all features that implies A by itself, regardless of the meaning of their interaction). This is not a desired case as long as the paper specified the interactions in section 3. Maybe it requires more detailed explanation about how good applying NID for this task is.

Experiments are conducted to show the behavior of the interaction detector and actual improvement of utilizing extracted interactions as additional features. Experiments look less informative for comparing the proposed method with other existing methods for similar motivations (e.g., some methods introduced in related work) since there is only a trivial baseline by the original LIME's method.

In the case-study of Figure 4(b), I thought that the proposed method is a bit biased for frequent but meaningless features, because it detected "I" or "a" that intuitively occur with any labels. It is maybe because the Algorithm 1. that simply aggregates all detected interactions regardless of their actual importance. For further improvement, it may be necessary to introduce some weighting strategy to detected interactions.


**Experience Assessment:**

I do not know much about this area.

**Review Assessment: Checking Correctness Of Derivations And Theory:**

I assessed the sensibility of the derivations and theory.

**Review Assessment: Checking Correctness Of Experiments:**

I assessed the sensibility of the experiments.

**Review Assessment: Thoroughness In Paper Reading:**

I read the paper at least twice and used my best judgement in assessing the paper.

---

> ### Author Response · Authors · 2019-11-13
> **Author Response**
>
> Thank you for your comments and observations. We address them below:
>
> >> “it is sensitive not only the true interaction between features but also set of features holding similar signals”
>
> This concern is actually one of the advantages of our proposal to use interaction detection and LIME together. The features generated by LIME are randomly sampled (Section 4.1), so they are uncorrelated by default. Therefore, NID focuses on identifying interaction effects rather than correlations. We added a comment on this in Section 4.1.
>
> >> “Experiments look less informative for comparing the proposed method with other existing methods for similar motivations (e.g., ...”
>
> We have added experiments in Appendix E comparing MADEX to baselines with similar motivations on local interpretation of feature subsets. MADEX performs well compared to the baselines, especially on interpreting neural network models.
>
> >> “In the case-study of Figure 4(b), I thought that the proposed method is a bit biased for frequent but meaningless features, because it detected "I" or "a"... it may be necessary to introduce some weighting strategy to detected interactions”
>
> We have several responses to this comment. 1) words such as “a” and “I” are often “stop words” in natural language processing, and it is standard practice to omit them [1,2]. 2) The interactions outputted by MADEX are actually ranked/weighted in our experiments, based on the usage of top-k thresholding in Section 5.3.1.
>
> To address your comment, we provide results in Appendix C.1 on randomly selected sentences excluding stop words, and we only show top-ranked word interactions. The interactions are meaningful for sentiment, such as (family, oriented), and (inarticulate, disappointing). In addition, we ran a more thorough interaction occurrence experiment in Appendix C.2 and found that the frequent word interactions are especially meaningful, i.e. (well, worth), (too, bad), (pretty, good), etc.
>
>
> [1] Christopher D. Manning, Prabhakar Raghavan, and Hinrich Schutze. Introduction to Information Retrieval. Cambridge University Press, 2008.
> [2] Jure Lescovec, Anand Rajaraman, Jefferey Ullman. Mining of Massive Datasets. Cambridge University Press, 2014.

---

### Official Review · AnonReviewer2 · 2019-11-02
**Official Blind Review #2**

**Rating:** 6

**Review:**

This paper is focused on identifying/discovering feature interactions in blackbox models (with a focus on recommendations). Specifically the technique works by first corrupting datapoints and then using this "local dataset" to find interacting features via lasso-regularized multi-level perceptron approach (NID from Tsang et al). Using an expanded feature space and repeated calls to the above steps this is then used to find the most commonly occurring patterns.

The empirical section is quite interesting, with some nice mix of qualitative and quantitative results. The quantitative results in particular are quite intriguing -- across recommendation and non-recommendation tasks.

Overall the paper makes for an interesting read. On the whole I lean slightly positive -- largely due to the empirical section and the quantitative gains observed by adding the feature interactions as features to the model.

That said I had quite a few concerns about the work:

- In general the exposition was quite lacking in the methodological section. I had to re-read the paper a few times to be able to make out to fully understand the underlying methodology. I would make the overall pipeline very clear and explain out the MADEX pipeline very clearly. The feature expansion section (4.3) in particular was not very clear -- as to how it fit in with the rest of the pipeline and something that could do with more work.  Nomenclature like source and target model are used somewhat arbitrarily and again can be clarified.

- Another concern I had with the approach is how this would work if the underlying features itself were not known. Say for example you had a text understanding model -- the tokenization and vocabulary/OOV etc .. may all be components of the blackbox. Likewise for vision models, what features are being used may vary.

By assuming (effectively perfect) information about the features used, this is no longer really a "blackbox" model. I would have wanted to see some deeper discussion on this topic.

I also had some concerns about the scalability of dense features and their bucketization in this approach. I'm not entirely convinced the method would work as well and efficiently in such feature spaces. I would have wanted to see some more empirical evidence and understanding on this topic.

If possible it would have also been great to understand higher-order interactions and the ability of the model to find them empirically.

- One thing I wasn't entirely certain of was the amount of novelty in the work since it leverages existing works like NID (Tsang et al) and LIME (Ribeiro et al) for some of the key aspects of the method. It would be nice to see some discussion on this front as well.

- I also would have liked to see significance testing being performed in general across the experiments.

- Another smaller concern stemmed from a lack of mention of the stopping criterion of the model. In particular I'm wondering how models were stopped further training / best checkpoint was picked. This would be good to elaborate on to make sure that the models were all compared fairly from a computational perspective.

- I also had some concerns with the scalability of the approach in large features spaces but am willing to give the authors the benefit of doubt on this one based on the 3 hour number they quoted from their experiments (which indicates something that is not prohibitively slow)


**Experience Assessment:**

I have read many papers in this area.

**Review Assessment: Checking Correctness Of Derivations And Theory:**

I assessed the sensibility of the derivations and theory.

**Review Assessment: Checking Correctness Of Experiments:**

I assessed the sensibility of the experiments.

**Review Assessment: Thoroughness In Paper Reading:**

I read the paper at least twice and used my best judgement in assessing the paper.

---

> ### Author Response · Authors · 2019-11-13
> **Author Response**
>
> Thank you for your thorough comments and suggestions. Our responses to your points are below:
>
> >> “In general the exposition was quite lacking in the methodological section”
>
> We revised the methodology section to clarify MADEX and the input/output of Section 4.3. We now use source/target nomenclature consistently through the paper.
>
> >> “how this would work if the underlying feature itself were not known”
>
> Did you mean if we can use GLIDER for text or image models? We did not claim that global interaction detection or encoding could work for these models, hence the “R” for recommendation in GLIDER. Nonetheless, we added preliminary experiments in Appendix C.2 showing that MADEX frequently detects meaningful word interactions from the Sentiment-LSTM model across multiple sentences, i.e., (well, worth), (too, bad), (pretty, good), etc. Doing the same for image models is very challenging and left for future work.
>
> >> “By assuming (effectively perfect) information about the features used, this is no longer really a "blackbox" model”
>
> There are many different models that use dense and sparse features, especially in recommendation. One can see this variety of model architectures already in literature [1-5]. In real-world commercial settings, recommender architectures can also be heavily engineered; therefore, it is useful to have methods that treat the recommender system as a black-box model. We made a note of our meaning of “black-box recommender model” in the notations section.
>
> >> “about the scalability of dense features and their bucketization”
>
> We added scalability experiments on dense feature bucketization in Appendix B. Our requirement that a valid cross feature ID occurs more than $T$ times (Section 4.3) restricts the growth in parameters as the number of buckets increases.
>
> >> “amount of novelty in the work”
>
> There is very limited work on improving prediction performance by model interpretations, especially in the automatic way we propose. In this perspective, our work takes a step towards bridging interpretability with the field of AutoML via automatic feature engineering. Moreover, this work exposes feature interactions learned by general prediction models, which is a crucial step for interpretability applications such as scientific discovery (e.g., identifying DNA sequence interactions (Section 5.3.2)) and trustworthy machine learning (e.g., understanding how ads are targeted to users via ads-user interactions).
>
> >> “to understand higher-order interactions and the ability of the model to find them empirically”
>
> We added frequencies of higher-order interactions detected from recommender, image, and text models in Appendix F. Higher-order interactions are common, especially from the image model.
>
> >> “significance testing being performed in general”
>
> We ran significance testing. For the global interactions in Figure 2, we test the significance of the top-1 occurrence counts using order statistics. For the Criteo dataset, any count > 43 is significant (p value < 0.05). Ours was 572. For the Avazu dataset, any count > 33 is significant. Ours was 423.
>
> >> “lack of mention of the stopping criteria of the model”
>
> Models are early stopped and selected based on best performance on validation sets. The Experiment Setup section mentioned early stopping.
>
>
> [1] Cheng, et al. Wide & deep learning for recommender systems. In Proc of the 1st workshop on deep learning for recommender systems, pp. 7–10. ACM, 2016.
> [2] Guo, et al. Deepfm: a factorization machine based neural network for ctr prediction. In Proc of IJCAI. 2017.
> [3] Wang, et al. Deep & cross network for ad click predictions. In Proc of the ADKDD’17, 2017.
> [4] Lian, et al. xdeepfm: Combining explicit and implicit feature interactions for recommender systems. In Proc of SIGKDD, 2018.
> [5] Song, et al. Autoint: Automatic feature interaction learning via self-attentive neural networks. In Proc of CIKM, 2019.

---

### Decision · Program_Chairs · 2019-12-19

**Decision:**

Accept (Poster)

**Comment:**

The paper extracts feature interactions in recommender systems and studies the effect of these interactions on the recommendations. While the focus is on recommender systems the authors claim that the ideas can be generalised to other domains also.

All reviewers found the empirical results and analysis thereof to be very interesting and useful. This paper saw a healthy discussion between the authors and reviewers and all reviewers agreed that this paper makes a useful contribution. I recommend that the authors address all the concerns of the reviewers in the final version of the paper.